# Predictive Validity of a New Triage System for Outcomes in Patients Visiting Pediatric Emergency Departments: A Nationwide Study in Korea

**DOI:** 10.3390/children10060935

**Published:** 2023-05-25

**Authors:** Woori Bae, Arum Choi, Chun Song Youn, Sukil Kim, Kyu Nam Park, Kyunghoon Kim

**Affiliations:** 1Department of Emergency Medicine, Seoul St. Mary’s Hospital, College of Medicine, The Catholic University of Korea, Seoul 06591, Republic of Korea; baewool7777@hanmail.net (W.B.); ycs1005@catholic.ac.kr (C.S.Y.); emsky@catholic.ac.kr (K.N.P.); 2Department of Preventive Medicine and Public Health, College of Medicine, The Catholic University of Korea, Seoul 06591, Republic of Korea; dyemelody@gmail.com (A.C.);; 3Department of Pediatrics, Seoul National University College of Medicine, Seoul 03080, Republic of Korea

**Keywords:** triage, child, emergency department, validity, hospitalization, intensive care unit

## Abstract

Triage is essential for rapid and efficient intervention in patients visiting an emergency department. In Korea, since 2016, the Pediatric Korean Triage and Acuity Scale (PedKTAS) has been implemented nationwide for the triage of patients visiting pediatric emergency departments (PEDs). The aim of this study was to evaluate the validity of the PedKTAS in patients who visit PEDs. This study was a retrospective observational study of national registry data collected from all emergency medical centers and institutions throughout Korea. We analyzed data from patients aged <15 years who visited emergency departments nationwide from January 2016 to December 2019. The hospitalization and intensive care unit (ICU) admission rates were analyzed on the basis of triage level. In total, 5,462,964 pediatric patients were included in the analysis. The hospitalization rates for PedKTAS Levels 1–5, were 63.5%, 41.1%, 17.0%, 6.5%, and 3.7%, respectively, and were significantly different (*p* < 0.001). The ICU admission rates for PedKTAS Levels 1–5 were 14.4%, 6.0%, 0.3%, 0.1%, and 0.1%, respectively, and were significantly different (*p* < 0.001). The hospitalization and ICU admission rates were highest for PedKTAS Level 1, and differences were significant based on the level. We identified that the PedKTAS is suitable for predicting the emergency status of pediatric patients who visit PEDs.

## 1. Introduction

A triage system that classifies the emergency status of a patient when visiting the emergency department is essential to provide preferential emergency treatment to patients [1]. In addition, triage systems can ameliorate inefficient operation and resource use caused by overcrowding in emergency departments. Therefore, various triage tools are used in many countries. The Emergency Severity Index [2], Manchester Triage Scale [3], and Canadian Emergency Department Triage and Acuity Scale (CTAS) [4] are the most widely used tools.

In 2001, the Canadian Pediatric Triage and Acuity Scale (PedCTAS) [5], a triage tool applied to pediatric patients, was developed in Canada and is used in most pediatric emergency departments (PEDs) in Canada. By using this tool, a nurse selects one of 38 main symptoms and then assigns one of five triage levels by synthesizing additional information about the patient, such as vital signs, baseline status, or age.

In Korea, the problem of inefficiency of the emergency medical system because of chronic overcrowding of PEDs has been continually raised. Before 2015, different triage systems were applied to children in each emergency department nationwide. Each emergency department classified patients’ urgency into three levels by applying ESI, classified into five levels by applying the PedCTAS, and some emergency departments did not even apply the triage system. For this reason, it was difficult to share information about patients’ emergency levels among hospitals or among 119 emergency medical services and medical staff at the pre-hospital stage. As a result, in 2010, an incident occurred in which a child with intussusception died because the emergency department of several hospitals did not recognize the emergent situation of the patient. Given these circumstances, a nationally unified triage system was needed to evaluate the degree of urgency in pediatric emergency patients. The Korean Society of Emergency Medicine (KSEM) developed the Pediatric Korean Triage and Acuity Scale (PedKTAS) [6] in 2015 by modifying and supplementing the PedCTAS to suit the situation in Korea because the PedCTAS was known as a triage tool whose reliability and validity were verified. The PedKTAS has been applied to pediatric patients visiting nationwide PEDs since 2016.

In the past few years, the PedKTAS has been implemented. However, studies on the validity of the PedKTAS have not yet been reported, and whether the PedKTAS is appropriate for predicting the clinical outcomes in patients visiting PEDs is unknown. The aim of this study was to identify the hospitalization rate and intensive care unit (ICU) admission rate on the basis of triage level after applying the PedKTAS and to compare them with the results of applying PedCTAS.

## 2. Materials and Methods

### 2.1. Study Setting, Design, and Population

This study was a retrospective, observational study. The data used in this study were retrieved from the National Emergency Department Information System (NEDIS). The NEDIS was established in 2003 as a nationwide emergency medical network in South Korea. Since 1 January 2016, all 31 district emergency medical centers (EMCs) in Korea, 120 regional EMCs, and 257 of 262 regional emergency medical institutions (EMIs) have participated and transmitted data to the NEDIS in real time. District EMCs are emergency centers assigned by the Minister of Health and Welfare to provide final care for critical emergency patients at tertiary general hospitals or general hospitals with more than 300 beds and to prepare for disaster situations. The average number of visits to District EMCs per year is approximately 40,000, and the average number of pediatric patients is approximately 12,000 per year. Regional EMCs are assigned by the provincial governor if they are suitable for emergency patient care among general hospitals; the average number of visits per year is approximately 30,000, and the average number of pediatric patients is approximately 8000 per year. Regional EMI is assigned by the mayor among general hospitals; the average number of patients visiting each year is approximately 6000, and the average number of pediatric patients is approximately 3000 per year.

Data for 4 years—from 1 January 2016 to 31 December 2019—were provided by the NEDIS. Patients younger than 15 years of age who visited the emergency department were included. Patients with KTAS 8 and 9 who were not classified by the PedKTAS or who visited a PED for certificate issuance were excluded from the analysis.

We considered each PED visit from arrival to departure as an independent case. The following variables were extracted from the NEDIS data during the study period: demographic variables, triage level, shift of arrival, EMC class, arrival mode, diagnosis at the PED, and final disposition. The demographic variables included age and sex. We divided the participants into four age groups: <1 year (i.e., infant), 1–4 years (i.e., young child), 5–9 years (i.e., preschool- and early-school-aged child), and 10–14 years (i.e., school-aged child). Diagnoses were determined by using the diagnostic codes of the Korean Standard Classification of Diseases-7 and the Korean version of the International Classification of Diseases-10th Revision [7,8]. “Hospitalization” included patients directly admitted to general wards, ICUs, operating rooms, and patients transferred to other hospitals for hospitalization. “ICU admission” was defined as direct admission from the PED to the ICU, admission to an ICU from an operating room, or transfer to another hospital for ICU admission.

The PedKTAS was developed by consensus of experts from KSEM. The PedKTAS is applied to children under the age of 15, and patients are divided into 17 symptom groups. In total, 166 specific symptoms are included in each symptom group, so the trained nurses can select the symptoms corresponding to the patient. In the last step, the final PedKTAS level is determined by selecting the appropriate item from the primary or secondary considerations (consciousness, hemodynamic signs, degree of dyspnea, body temperature, hemorrhagic disease, immune state, and accident mechanism) to evaluate the severity and urgency of the symptom. A screenshot of the process of implementing the PedKTAS is provided in Appendix A. The priority of pediatric patient care is determined according to the degree of emergency, and if they are classified as Levels 1 or 2 in the first impression evaluation, they receive treatment with the highest priority. For other patients, the treatment area may change depending on whether an infectious disease is suspected, and the priority of treatment is determined according to the severity. In this process, if the severity is low, the waiting time may be long. Re-evaluation is recommended within 30 min, 1 h, and 2 h for Levels 3, 4, and 5, respectively, in order to evaluate whether the waiting patient’s symptoms worsened. There are differences between the two triage tools in terms of application age, chief complaint list, and criteria for abnormal vital signs. Details of the comparison between the PedKTAS and the PedCTAS are described in Table 1.

### 2.2. Outcomes

The primary outcomes were the hospitalization and ICU admission rates based on the PedKTAS level. The secondary outcome was to compare the hospitalization and ICU admission rates with application of the PedKTAS and the PedCTAS, respectively.

### 2.3. Statistical Analysis

Continuous variables were expressed as the median and interquartile range, and categorical variables were expressed as the frequency and percentage. The PedKTAS level was treated as an independent variable, and the hospitalization and ICU admission rates were treated as the dependent variables. The hospitalization and ICU admission rates were compared using the odds ratios based on the PedKTAS level; a logistic regression analysis was conducted at 95% reliability, adjusted for age, sex, mode of arrival, and diagnosis. All analyses were conducted using R version 4.0.0 (R Foundation for Statistical Computing, Vienna, Austria). Because the sample size of the analysis data was very large, the probability level of significance was set at *p* < 0.001.

## 3. Results

From January 2016 to December 2019, 6,847,960 patients were extracted by using the NEDIS. Among these, 1,322,689 patients without a triage classification and 62,307 patients with missing data were excluded. Ultimately, 5,462,964 patients were analyzed in this study (Figure 1).

On the basis of the NEDIS data, the 1- to 4-year-old age group constituted the highest proportion of patients (48.1%), followed by the 5- to 9-year-old age group (24.9%). Boys accounted for 57.2% of the total participants. The proportions of patients were 0.2%, 3.4%, 31.6%, 55.8%, and 9.0% for PedKTAS Levels 1, 2, 3, 4, and 5, respectively. The demographic data of the patients in the study group are summarized in Table 2.

The hospitalization rates were 63.5%, 41.1%, 17.0%, 6.5%, and 3.7% for PedKTAS Levels 1, 2, 3, 4, and 5, respectively, and were significantly different (*p* < 0.001). The ICU admission rates were 14.4%, 6.0%, 0.3%, 0.1%, and 0.1% for PedKTAS Levels 1, 2, 3, 4, and 5, respectively, and were significantly different. Deaths in the PED were 0.6% of 75 patients at PedKTAS Level 1, and 47 patients, 29 patients, and 9 patients at PedKTAS Levels 2, 3, and 4, respectively. The emergency department length of stay (LOS) was the longest at PedKTAS Level 1 at 268.8 min, and the emergency department LOS decreased from PedKTAS Level 1 to Level 5 (Table 3).

The PedKTAS Level 1 in the trauma group had the highest ICU admission rate at 29.5%. Regarding medical diseases, ICU admission rates were the highest in the cardiac disease group at PedKTAS Level 1 and Level 2, at 26.4% and 14.4%, respectively. The results of hospitalization rate and ICU admission rate according to PedKTAS Level for each disease group are shown in Table 4.

In this study, the hospitalization rates based on the PedKTAS were compared with the expected hospitalization rates proposed by the PedCTAS (i.e., Level 1, 80%; Level 2, 55%; Level 3, 30%; Level 4, 15%; and Level 5, 5%) and the hospitalization rates in a multicenter study that applied the PedCTAS (i.e., Level 1, 61%; Level 2, 30%; Level 3, 10%; Level 4, 2%; and Level 5, 0.9%) [9,10]. At all levels, the hospitalization rate based on the PedKTAS level was lower than the hospitalization rate proposed by the PedCTAS but higher than the hospitalization rate of the multicenter study [9] that applied PedCTAS (Figure 2A). The ICU admission rates based on PedKTAS levels were compared with the ICU admission rates in a multicenter study [9] that applied the PedCTAS (i.e., Level 1, 24.0%; Level 2, 2.5%; Level 3, 0.1%; Level 4, 0%; and Level 5, 0%). The ICU admission rate at Level 1 was higher in the PedCTAS than in the PedKTAS, but the ICU admission rate at Level 2 was higher in the PedKTAS than in the PedCTAS (Figure 2B).

Figure 3A shows the hospitalization rates by triage level based on the hospital type. In district EMCs, the hospitalization rates based on triage level were 67.2%, 44.5%, 19.1%, 7.5%, and 4.6% for PedKTAS Levels 1, 2, 3, 4, and 5, respectively. In the regional EMCs, the hospitalization rates based on triage level were 61.0%, 38.9%, 16.5%, 6.4%, and 3.7% for PedKTAS Levels 1, 2, 3, 4, and 5, respectively. In regional EMIs, the hospitalization rates based on triage levels were 43.6%, 26.1%, 9.3%, 4.9%, and 2.3% for PedKTAS Levels 1, 2, 3, 4, and 5, respectively.

Figure 3B shows the ICU admission rates by triage level based on hospital type. In district EMCs, the ICU admission rates based on the triage level were 17.9%, 7.4%, 0.4%, 0.1%, and 0.2% for PedKTAS Levels 1, 2, 3, 4, and 5, respectively. In regional EMCs, the ICU admission rates based on triage level were 11.8%, 4.9%, 0.3%, 0.1%, and 0.1% for PedKTAS Levels 1, 2, 3, 4, and 5, respectively. In regional EMIs, the ICU admission rates based on the triage level were 0%, 3.0%, <0.1%, <0.1%, and <0.1% for PedKTAS Levels 1, 2, 3, 4, and 5, respectively.

## 4. Discussion

The emergency triage system, which determines patients’ severity, is essential for the priority treatment of emergency patients and the efficient distribution of medical resources. This study is the first validation study conducted after the implementation of the PedKTAS in Korea. The results of this study will provide sufficient evidence for predicting the emergency status of patients in PEDs.

In this study, we found that when the PedKTAS was applied to patients visiting PEDs, the hospitalization and ICU admission rates were the highest among patients in PedKTAS Level 1. We also found that the hospitalization and ICU admission rates significantly decreased as the severity of the PedKTAS level decreased from 1 to 5. These results indicated that the PedKTAS can reflect differences in patients’ statuses, depending on their level. On the basis of this study, it can be said that the higher the PedKTAS level from 5–1, the more severe the patient’s condition, and the faster the examination and treatment that are needed. Previous studies on the validity of the PedCTAS, which were conducted on the basis of the PedKTAS, also reported similar results. An Israeli study [11] reported that when the PedCTAS was applied, as the severity of the PedCTAS level increased from 5 to 1, a strong correlation existed between the hospitalization rate and ICU admission rate. Another study [12] conducted in Costa Rica also reported that, as the severity of the PedCTAS level increased, the hospitalization rate and ICU admission rate increased.

In a multicenter study [9] using the PedCTAS, the hospitalization rates based on triage level were 61%, 30%, 10%, 2%, and 0.9% for Levels 1–5, respectively [9]. These results were lower than the expected hospitalization rates based on PedCTAS at all levels except for Level 5. In the current study, the hospitalization rates for PedKTAS Levels 2 and 5 were 41.1% and 3.7%, respectively, which was within the expected hospitalization rate and lower than the expected hospitalization rate, respectively, of the other levels.

However, compared with the findings of a previous multicenter study, [9] the current study result was closer to the expected hospitalization rates. The reason for the lower-than-expected hospitalization rate was likely related to over-triage. When triaging pediatric patients, the important criteria are the vital signs and whether the patient appears unwell. However, when pediatric patients visit PEDs with fear, their irritability worsens, and their condition may be evaluated as more severe than it actually is [13]. Another study [14] reported no difference in the hospitalization rates of children who visited PEDs because of fever within 24 h when the triage level was lower than the actual triage level. This over-triage may explain why the actual hospitalization rate was lower than expected.

The ICU admission rate based on triage level differed between the PedKTAS and the PedCTAS. The ICU admission rate at Level 1 was lower for the PedKTAS than for the PedCTAS, but the ICU admission rate at Level 2 was higher for the PedKTAS than for the PedCTAS. The PedCTAS multicenter study included 12 tertiary care university-affiliated hospitals with ICUs; therefore, admitting Level 1 patients to the ICU may have been easier. However, this study analyzed nationwide data and included hospitals of all sizes. Thus, the ICU admission of Level 1 patients would be restricted in a tertiary hospital without a pediatric ICU. However, in a secondary hospital with an ICU, admitting Level 1 patients and Level 2 patients to the ICU would be easier. In Korea, the ICU admission rate of PedKTAS Level 1 patients in a single tertiary care university-affiliated hospital was 34% [15]. The fact that the ICU admission rate of PedKTAS Level 1 patients is lower than that of PedCTAS Level 1 patients has important clinical significance. The PedKTAS may be overestimating the severity and urgency of pediatric patients. It may be better to overestimate than underestimate the condition of pediatric patients. However, in places where the use of medical resources is limited, such as in an emergency department, too many Level 1 patients requiring prompt treatment may be screened, and treatment of patients actually requiring emergency treatment may be delayed. Therefore, it may be necessary to modify the evaluation items for selecting PedKTAS Level 1 patients.

In this study, hospitalization and ICU admission rates were identified via the PedKTAS level based on hospital type through subgroup analysis. District EMCs had the highest hospitalization rate at all levels, followed by regional EMCs and EMIs. The ICU admission rate also showed a similar trend in the hospitalization rate based on hospital type. This finding is likely because higher-level hospitals are equipped with specialized medical staff and facilities to accommodate many patients. A previous study [16] in the United States also reported that tertiary hospitals had a higher hospitalization rate because of the availability of 24 h specialist care, advanced testing, and other services that are unavailable in small hospitals. We identified that PedKTAS Level 1 patients were not admitted to ICUs in regional EMIs, likely because small hospitals, such as regional EMIs, have several restrictions on the treatment of Level 1 pediatric patients, such as lack of specialists or equipment. Therefore, they are transferred to higher-level hospitals.

The triage of pediatric patients visiting the emergency department is performed by nurses. Nurses who perform triage using the PedKTAS must complete 6 h of training [17]. The educational process consists of basic learning lectures on severity and urgency classification, group discussions on severity classification cases, and written tests. Nurses who completed this training course showed a considerable level of reliability in severity classification using the PedKTAS. One study reported a weighted kappa value of 0.76 for inter-nurse agreement [18]. This result shows that the range of weighted kappa values identified in the meta-analysis study of inter-rater agreement of the PedCTAS is 0.41 to 0.74, which is significant [19].

This study had several limitations. First, the data for this study were provided anonymously. Therefore, information on the same patient is likely to have been included more than once. However, this factor is an inherent limitation of studies with anonymized data and is not specific to this study. Second, this study analyzed the triage level results classified in actual clinical practice. Therefore, conducting in-depth analysis, such as inter-rater reliability analysis, was not possible. However, this inherent limitation is unavoidable in retrospective observational studies. Third, 1,309,867 patients not classified by the PedKTAS were excluded from this study. They accounted for approximately one-fifth of the total number of pediatric patients analyzed in this study, and most visited local EMIs. This was identified to have occurred in the early days of implementing the PedKTAS nationwide. Due to this, there is a possibility that sample selection bias exists. In the future, it is thought that additional analysis will be needed by extending the study period except for the initial period of KTAS implementation. Finally, this study reflected the medical situation in Korea; the results may differ depending on the medical situation and the environment in other countries.

The emergency department has the characteristics that the number of patients visited and the condition cannot be predicted, and treatment must be performed with limited medical resources. Therefore, overcrowding in emergency departments is an important problem that is directly related to the threat to patient safety. Efforts are being made in various ways to relieve the overcrowding of emergency departments. Using the triage system, the flow of patients can be controlled in the pre-hospital stage, and the patient’s condition can be accurately evaluated at the initial stage so that patients visiting the emergency department can wait safely in the hospital stage. Through this, overcrowding in the emergency department can be improved. Continuous efforts and research of many researchers will be required for the development and performance improvement of the triage system.

## 5. Conclusions

The hospitalization and ICU admission rates based on PedKTAS levels were the highest for PedKTAS Level 1 and were significantly different based on the level. We identified that the PedKTAS is suitable for predicting the emergency status of pediatric patients who visit PEDs.

## Figures and Tables

**Figure 1 children-10-00935-f001:**
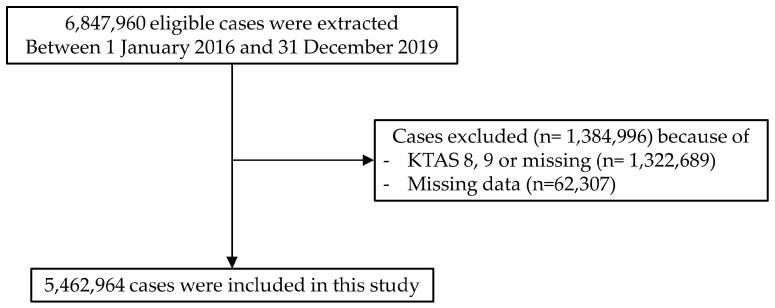
Flowchart of the study population.

**Figure 2 children-10-00935-f002:**
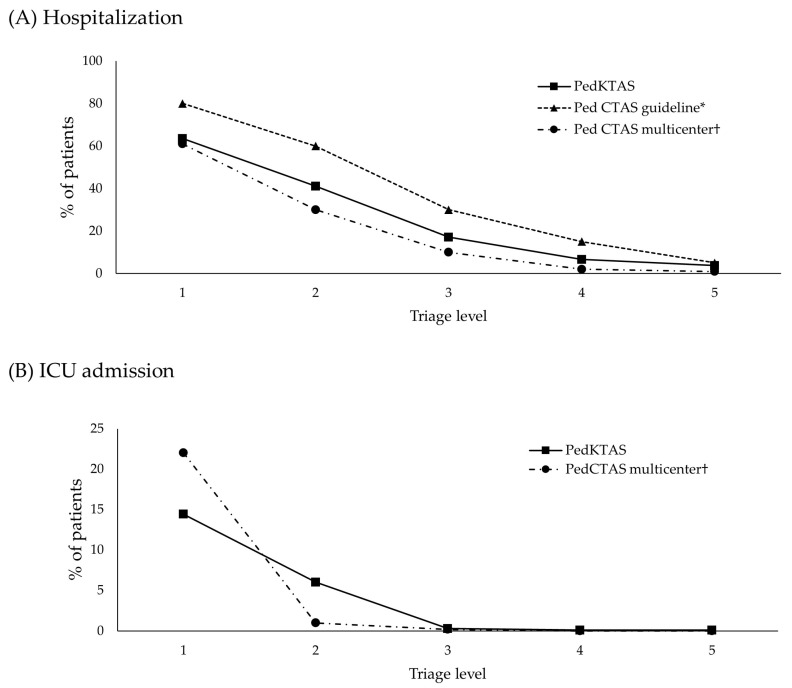
Hospitalization and ICU admission rates according to triage levels. (**A**) Comparison of hospitalization rates according to triage levels. (**B**) ICU admission rate according to triage levels. PedKTAS: Pediatric Korean Triage and Acuity Scale, PedCTAS: Pediatric Canadian Triage and Acuity Scale, ICU: intensive care unit. * Data from Gravel et al. [9]; † Data from Gravel et al. [10].

**Figure 3 children-10-00935-f003:**
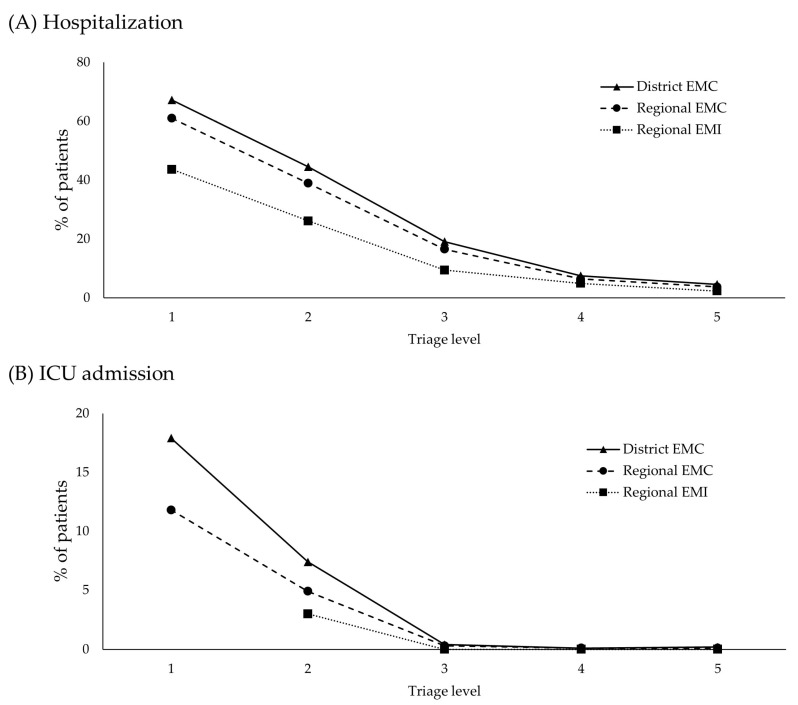
Hospitalization and ICU admission rates by triage level according to the hospital type. (**A**) Comparison of hospitalization rates according to triage levels. (**B**) ICU admission rate according to triage levels. EMC: emergency medical center, EMI: emergency medical institution, ICU: intensive care unit.

**Table 1 children-10-00935-t001:** Comparison of Pediatric Korean Triage and Acuity Scale and Pediatric Canadian Triage and Acuity Scale characteristics.

Criteria	PedKTAS	PedCTAS
Time to triage assessment	Immediately	10 min
Applicable age	<15 years old	<18 years old
Time to nurse reassessment	Immediately/15/30/60/120 min	Immediate/15/30/60/120 min
Triage decision criteria	Complaints based on 1st/2nd modifiers used	Complaints based on 1st/2nd modifiers used
Chief complaint list	17 categories (166 chief complaints)	18 categories (161 chief complaints)
Criteria for vital signs	Further segmentation by age (every 3 months under 3 years old, every year over 3 years old)	Organized by age group (every 3 months for children under 1 year old, 1–3 years old, 4–6 years old, 7–10 years old)
Range of abnormal vital signs	Wider	Narrower
Education method	Online or offline training	Web-based training

PedKTAS: Pediatric Korean Triage and Acuity Scale, PedCTAS: Pediatric Canadian Triage and Acuity Scale.

**Table 2 children-10-00935-t002:** Characteristics of patients visiting the pediatric emergency department.

Variables	Participants (n = 5,462,964)
Age, y	
<1	649,823 (11.9%)
1–4	2,629,488 (48.1%)
5–9	1,358,398 (24.9%)
10–14	825,255 (15.1%)
Gender	
Male	3,123,923 (57.2%)
Triage level	
KTAS 1	11,547 (0.2%)
KTAS 2	185,748 (3.4%)
KTAS 3	1,725,375 (31.6%)
KTAS 4	3,049,212 (55.8%)
KTAS 5	491,082 (9.0%)
Shift of arrival	
Day	1,390,799 (25.5%)
Evening	3,031,962 (55.5%)
Night	1,040,203 (19.0%)
Mode of arrival	
Self-referred	5,070,310 (92.8%)
Outpatient department	42,416 (0.8%)
Referred from clinic	350,238 (6.4%)
Diagnosis	
Medical	3,841,425 (70.3%)
Trauma	1,621,539 (29.7%)
Final disposition	
Discharged	4,864,771 (89.1%)
Hospitalization	
Admission to GW	536,582 (9.8%)
Admission to ICU	21,447 (0.4%)
Admission to OR	15,202 (0.3%)
Transfer to other hospital	23,490 (0.4%)
Discharged	4,864,771 (89.1%)
Expired	150 (0.0%)
DOA	1312 (0.0%)

Values are presented as *n* (%) unless otherwise indicated. KTAS: Korean Triage and Acuity Scale, GW: general ward, ICU: intensive care unit, OR: operation room, DOA: dead on arrival.

**Table 3 children-10-00935-t003:** Hospitalization and ICU admission according to triage levels in patients visiting the pediatric emergency department.

Variables	KTAS 1 (n = 11,547)	KTAS 2 (n = 185,748)	KTAS 3 (n = 1,725,375)	KTAS 4 (n = 3,049,212)	KTAS 5 (n = 491,082)
Hospitalization					
Admission to GW	4189 (36.3%)	60,481 (32.6%)	270,846 (15.7%)	184,170 (6.0%)	16,896 (3.4%)
Admission to ICU	1664 (14.4%)	11,187 (6.0%)	5709 (0.3%)	2267 (0.1%)	620 (0.1%)
Admission to OR	138 (1.2%)	744 (0.4%)	7508 (0.4%)	6533 (0.2%)	279 (0.1%)
Transfer to other hospital	1341 (11.6%)	3929 (2.1%)	10,306 (0.6%)	7392 (0.2%)	522 (0.1%)
Discharged	2846 (24.6%)	109,351 (58.9%)	1,430,974 (82.9%)	2,848,840 (93.4%)	472,760 (96.3%)
Expired	75 (0.6%)	47 (0.0%)	29 (0.0%)	9 (0.0%)	0 (0.0%)
DOA	1294 (11.2%)	9 (0.0%)	3 (0.0%)	1 (0.0%)	5 (0.0%)
Mean ED LOS, min (95% CI)	268.8 (255.8–281.7)	224.0 (222.1–225.8)	157.2 (156.8–157.6)	106.4 (106.1–106.7)	63.8 (63.4–64.2)
Hospitalization rate	46.17 (44.32–48.11)	14.80 (14.54–15.06)	5.32 (5.24–5.40)	1.88 (1.85–1.91)	1
* *p*-value	<0.001	<0.001	<0.001	<0.001	
ICU admission rate	137.56 (124.87–151.55)	24.62 (22.68–26.73)	2.41 (2.22–2.62)	0.67 (0.61–0.73)	1
* *p*-value	<0.001	<0.001	<0.001	<0.001	

Values are presented as *n* (%) or odds ratio (95% confidence interval). ** p*-values are based on multiple logistic regression compared with KTAS 5. KTAS: Korean Triage and Acuity Scale, GW: general ward, ICU: intensive care unit, OR: operation room, DOA: dead on arrival, ED LOS: emergency department length of stay.

**Table 4 children-10-00935-t004:** Regression analysis according to triage levels by disease group of patients visiting pediatric emergency department.

Variables	KTAS 1	KTAS 2	KTAS 3	KTAS 4	KTAS 5
Infectious disease	*n =* 185	*n =* 23,357	*n =* 325,241	*n =* 423,318	*n =* 37,630
Hospitalization (%)	85 (46.0)	11,948 (51.2)	45,326 (13.9)	37,426 (8.8)	2435 (6.5)
OR (95% CI)	12.47 (9.28–16.76)	10.41 (9.91–10.94)	2.37 (2.27–2.47)	1.51 (1.44–1.57)	Reference
*p*-value	<0.001	<0.001	<0.001	<0.001	
ICU admission (%)	23 (12.4)	1289 (5.5)	502 (0.2)	368 (0.1)	52 (0.1)
OR (95% CI)	142.7 (77.45–262.95)	11.51 (8.71–15.22)	1.14 (0.86–1.52)	0.99 (0.74–1.33)	Reference
*p*-value	<0.001	<0.001	0.3569	0.9579	
Respiratory disease	*n =* 827	*n =* 45281	*n =* 585874	*n =* 581262	*n =* 78493
Hospitalization (%)	569 (68.8)	14,100 (31.1)	88,744 (15.2)	56,629 (9.7)	7212 (9.2)
OR (95% CI)	21.92 (18.87–25.47)	3.89 (3.77–4.02)	1.77 (1.72–1.81)	1.14 (1.11–1.17)	Reference
*p*-value	<0.001	<0.001	<0.001	<0.001	
ICU admission (%)	145 (12.4)	920 (2.0)	754 (0.1)	219 (0.1)	64 (0.1)
OR (95% CI)	296.27 (216.99–404.51)	17.69 (13.68–22.87)	1.75 (1.36–2.27)	0.58 (0.44–0.77)	Reference
*p*-value	<0.001	<0.001	<0.001	<0.001	
Gastrointestinal disease	*n =* 38	*n =* 2810	*n =* 81357	*n =* 148,683	*n =* 16,527
Hospitalization (%)	15 (39.5)	922 (32.8)	22,732 (27.9)	17,902 (12.0)	766 (4.6)
OR (95% CI)	11.29 (5.77–22.08)	8.49 (7.61–9.46)	7.08 (6.57–7.63)	2.59 (2.41–2.79)	Reference
*p*-value	<0.001	<0.001	<0.001	<0.001	
ICU admission (%)	4 (10.5)	173 (6.2)	171 (0.2)	81 (0.1)	7 (0.1)
OR (95% CI)	359.39 (92.91–1390.14)	65.75 (30.76–140.54)	6.77 (3.17–14.46)	2.03 (0.94–4.41)	Reference
*p*-value	<0.001	<0.001	<0.001	0.073	
Neurological disease	*n =* 2730	*n =* 5250	20839	*n =* 13112	*n =* 1248
Hospitalization (%)	2062 (75.5)	3322 (63.3)	10,066 (48.3)	4869 (37.1)	297 (23.8)
OR (95% CI)	9.7 (8.27–11.38)	5.09 (4.41–5.88)	3.27 (2.85–3.74)	2.16 (1.88–2.48)	Reference
*p*-value	<0.001	<0.001	<0.001	<0.001	
ICU admission (%)	384 (14.1)	363 (6.9)	293 (1.4)	38 (0.3)	9 (0.7)
OR (95% CI)	22.2 (11.4–43.21)	9.2 (4.73–17.88)	2.09 (1.07–4.06)	0.46 (0.22–0.95)	Reference
*p*-value	<0.001	<0.001	0.0306	0.0347	
Cardiac disease	*n =* 1279	*n =* 1443	*n =* 8165	*n =* 6602	*n =* 447
Hospitalization (%)	551 (43.1)	687 (47.6)	2464 (30.2)	1388 (21.0)	112 (25.1)
OR (95% CI)	1.73 (1.35–2.21)	2.71 (2.13–3.44)	1.45 (1.16–1.81)	0.91 (0.73–1.14)	Reference
*p*-value	<0.001	<0.001	<0.001	0.3996	
ICU admission (%)	338 (26.4)	208 (14.4)	246 (3.0)	41 (0.6)	7 (1.6)
OR (95% CI)	15.64 (7.31–33.48)	10.21 (4.76–21.9)	2.28 (1.07–4.88)	0.48 (0.21–1.08)	Reference
*p*-value	<0.001	<0.001	0.0336	0.0757	
Trauma	*n =* 579	*n =* 20,057	*n =* 225,811	*n =* 1,194,108	*n =* 135,774
Hospitalization (%)	356 (61.5)	4746 (23.7)	24,318 (10.8)	26,468 (2.2)	1836 (1.4)
OR (95% CI)	117.61 (98.38–140.59)	23.72 (22.41–25.12)	8 (7.63–8.40)	1.63 (1.56–1.71)	Reference
*p*-value	<0.001	<0.001	<0.001	<0.001	
ICU admission (%)	171 (29.5)	902 (4.5)	884 (0.4)	562 (0.1)	31 (0.1)
OR (95% CI)	1717.91 (1156.06–2552.83)	194.97 (136.27–278.96)	15.82 (11.06–22.62)	2.08 (1.45–2.99)	Reference
*p*-value	<0.001	<0.001	<0.001	<0.001	

Values are presented as *n* (%) or odds ratio (95% confidence interval); *p*-values are based on multiple logistic regression compared with KTAS 5; KTAS: Korean Triage and Acuity Scale, ICU: intensive care unit.

## Data Availability

Data are available on request owing to restrictions, such as privacy or ethics.

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
