# Peer review of "Predictive Validity of a New Triage System for Outcomes in Patients Visiting Pediatric Emergency Departments: A Nationwide Study in Korea"

_children, 2023, doi:10.3390/children10060935_

Round 1

Reviewer 1 Report

The authors aimed to analyze the validity of the Pediatric Korean Triage and Acuity Scale (PedKTAS) through a nationwide study in Korea. Although this multicenter study is retrospective, more than 5 million pediatric patients (<15y old) who visited emergency departments (ED) in Korea were included from 2016 to 2019, allowing the authors to comprehensively analyze the validity of the PedKTAS.

The topic is of interest because their results may have an impact on managing pediatric patients in Korea by organizing resources of the ED in Korea according to the triage level given by the PedKTAS, and, therefore, their care pathways.

Introduction

Well written, consisted, documented and referenced. The research question is well exposed and argued.

May the authors justify their need to use their own triage tool (PedKTAS) and not the others (PedCTAS, ESI, MTS)?

Materials and Methods:

-It would be helpful to describe briefly the organization of the emergency departments in Korea? Are the patients profiles the same regarding the type of emergency departments: District emergency medical centers (EMCs) versus regional EMCs versus regional emergency medical institutions (EMIs) (mix of adult and pediatric patients, how many pediatric visits per year)? What about their available resources according to the type of emergency (emergency physicians? Emergency pediatricians? ICU? PICU? pediatric hospitalization units? pediatric nurses, etc…)

-it would be valuable to present the PedKTAS: how was it build? (expert consensus? Which guidelines?) how many signs and symptoms? any algorithm? any diagnosis included in the tool? Weight of the vital signs in guiding the nurses to assign the triage level? Any specific profiles that can upgrade the triage level (i.e. new born, immunosuppression)? It would be useful to have a screenshot or a Table with the triage tool itself

-As other tools, it would be interesting to describe how the nurse are trained to use this tool? please consider and discuss aspects relating to implementation of the PedKTAS

-Because the sample size is huge, the p value set at .05 does not avoid random statistical significance (Wayant C, Scott J. Vassar M. Evaluation of lowering the P-value threshold for statistical significance from 05 to 005 in previously published randomized clinical trials in major medical journals. JAMA. (2018) 320:1813– 5. doi: 10.1001/jama.2018.12288). It would worth to set the p value set at .001 (or less).

Results

-please analyze validity of the PedKTAS to certain diseases (infections, etc).

-please provide data on severity outcomes such as mortality, length of stay

Discussion

-Authors compared the hospitalization and ICU rates based on PedKTAS levels, with those of the PedCTAS levels. Did the authors check if the study populations were comparable?

-Authors reported lower rates of hospitalization and ICU admissions due to the trend of over-triage of the PedKTAS tool. They argued that pediatric patients visit PEDs with fear, their irritability worsens and their condition may be evaluated as more severe than it actually is. What did the authors suggest to face those situations that are relatively frequent? Did the authors mean that PedKTAS is not sufficiently acute to triage those patients?

-In the future, will the authors plan to analyze the validity of their triage tool PedKTAS versus a gold standard? (i.e. PedCTAS)

Author Response

The topic is of interest because their results may have an impact on managing pediatric patients in Korea by organizing resources of the ED in Korea according to the triage level given by the PedKTAS, and, therefore, their care pathways.

Thank you for taking the time to review our manuscript and for providing meaningful comments. We have made corrections and clarifications in the revised manuscript after carefully reading your comments.

Introduction

Well written, consisted, documented and referenced. The research question is well exposed and argued.

May the authors justify their need to use their own triage tool (PedKTAS) and not the others (PedCTAS, ESI, MTS)?

Response: Thank you for your important note. Since I did not clearly describe the reason for choosing PedKTAS in Korea, I think it is something that readers might be curious about. This is explained in more detail in the Introduction section as follows.

Line 44-57

Before 2015, different triage systems were applied to children in each emergency department nationwide. Each emergency department classified patients' urgency into three levels by applying ESI, classified into five levels by applying PedCTAS, and even some emergency departments did not apply the Triage system. For this reason, it was difficult to share information about patients' emergency levels between hospitals or between 119 emergency medical services and medical staff at the pre-hospital stage. As a result, in 2010, an incident occurred in which a child with intussusception died because the emergency department of several hospitals did not recognize the emergent situation of the patient. Given these circumstances, a nationally unified triage system was needed to evaluate the degree of urgency in pediatric emergency patients. The Korean Society of Emergency Medicine (KSEM) developed the Pediatric Korean Triage and Acuity Scale (PedKTAS) [6] in 2015 by modifying and supplementing the PedCTAS to suit the situation in Korea because PedCTAS was known as a triage tool whose reliability and validity were verified.

Materials and Methods:

-It would be helpful to describe briefly the organization of the emergency departments in Korea? Are the patients profiles the same regarding the type of emergency departments: District emergency medical centers (EMCs) versus regional EMCs versus regional emergency medical institutions (EMIs) (mix of adult and pediatric patients, how many pediatric visits per year)? What about their available resources according to the type of emergency (emergency physicians? Emergency pediatricians? ICU? PICU? pediatric hospitalization units? pediatric nurses, etc…)

Response: Thank you for your important note of what we were missing. Although it is difficult to present all the size of individual hospital emergency departments, the number of patients visited, and the number of medical staff, it is possible to present the characteristics according to the size of emergency departments and the average number of patients visited. In the pediatric emergency department in Korea, there are pediatric specialists or pediatric residents alone, or emergency medicine specialists alone, and pediatrics and emergency medicine specialists alternately working in emergency departments. The average number of patients visited by pediatric emergency room size was added as follows.

Line 72-82

District EMCs are emergency centers assigned by the Minister of Health and Welfare to provide final care for critical emergency patients at tertiary general hospitals or general hospitals with more than 300 beds, and to prepare for disaster situations. The average number of visits to District EMCs per year is about 40,000 and the average number of pediatric patients is about 12,000 per year. Regional EMCs are assigned by the provincial governor if they are suitable for emergency patient care among general hospitals, and the average number of visits per year is about 30,000 and the average number of pediatric patients is about 8,000 per year. Regional EMI is assigned by the mayor among general hospitals, and the average number of patients visiting each year is about 6,000 and the average number of pediatric patients is about 3,000 per year.

-it would be valuable to present the PedKTAS: how was it build? (expert consensus? Which guidelines?) how many signs and symptoms? any algorithm? any diagnosis included in the tool? Weight of the vital signs in guiding the nurses to assign the triage level? Any specific profiles that can upgrade the triage level (i.e. new born, immunosuppression)? It would be useful to have a screenshot or a Table with the triage tool itself

Response: Thank you for your important note of what we were missing. To aid readers' understanding, I will add an explanation of how PedKTAS was developed and which algorithm evaluates the patient's urgency. The following explanation was added to the methods section. And the process of implementing PedKTAS is attached to Appendix A as a screenshot.

Line 100-115

PedKTAS was developed by consensus of experts from KSEM. PedKTAS is applied to children under the age of 15, and patients are divided into 17 symptom groups. Total 166 specific symptoms are included in each symptom group, so the trained nurses can select the symptoms corresponding to the patient. In the last step, the final PedKTAS level is determined by selecting the appropriate item from the primary or secondary considerations (consciousness, hemodynamic signs, degree of dyspnea, body temperature, hemorrhagic disease, immune state, accident mechanism) to evaluate the severity and urgency of the symptom. A screenshot of the process of implementing PedKTAS is attached to Appendix A. The priority of pediatric patient care is determined according to the degree of emergency, and if they are classified as Levels 1 or 2 in the first impression evaluation, they receive treatment with the highest priority. For other patients, the treatment area may change depending on whether an infectious disease is suspected, and the priority of treatment is determined according to the severity. In this process, if the severity is low, the waiting time may be long. It is recommended to re-evaluate within 30 minutes, 1 hour, and 2 hours for Level 3, 4, and 5, respectively, in order to evaluate whether the waiting patient's symptoms worsen.

Appendix A. Screenshots of the process of implementing Pediatric Korean Triage and Acuity Scale.

-As other tools, it would be interesting to describe how the nurse are trained to use this tool? please consider and discuss aspects relating to implementation of the PedKTAS

Response: Thank you for your important note. It is an important point to discuss how to train nurses who actually apply PedKTAS in emergency departments and whether it is reliable. In this regard, the following explanation has been added to the discussion section.

Line 267-275

The triage of pediatric patients visiting the emergency department is performed by nurses. Nurses who perform triage using PedKTAS must complete 6 hours of training [17]. The educational process consists of basic learning lectures on severity and urgency classification, group discussions on severity classification cases, and written tests. Nurses who completed this training course showed a considerable level of reliability in severity classification using PedKTAS. One study reported a weighted kappa value of 0.76 for inter-nurse agreement [18]. This result shows that the range of weighted-kappa values identified in the meta-analysis study of interrater agreement of PedCTAS is 0.41 to 0.74, which is significant [19].

[Reference]

  1. The Korean Society of Emergency Medicine. Korean Triage and Acuity Scale. Available online: http://www.ktas.org (accessed on May 12).
  2. Yang, J.; Lee, E. Inter-rater Reliability of Korean Triage and Acuity Scale (KTAS) among the Research Nurses and the Triage Nurses. Journal of East-West Nursing Research 2020, 26, 91-99, doi:10.14370/jewnr.2020.26.1.91.
  3. Mirhaghi, A.; Heydari, A.; Mazlom, R.; Ebrahimi, M. The Reliability of the Canadian Triage and Acuity Scale: Meta-analysis. N Am J Med Sci 2015, 7, 299-305, doi:10.4103/1947-2714.161243.

-Because the sample size is huge, the p value set at .05 does not avoid random statistical significance (Wayant C, Scott J. Vassar M. Evaluation of lowering the P-value threshold for statistical significance from 05 to 005 in previously published randomized clinical trials in major medical journals. JAMA. (2018) 320:1813– 5. doi: 10.1001/jama.2018.12288). It would worth to set the p value set at .001 (or less).

Response: Thank you for pointing this out. Because the sample size of the analysis data was very large, statistical significance was set to less than 0.001. As you said, it was revised and described as follows.

Line 131-132

Because the sample size of the analysis data was very large, the probability level of significance was set at P<0.001.

Results

-please analyze validity of the PedKTAS to certain diseases (infections, etc).

Response: Thank you for pointing this out. As you said, it would be very important to identify the validity according to PedKTAS for each disease group. By analyzing the validity of PedKTAS according to disease group, it will be possible to find out which disease group is more effective in severity classification. Analysis results are updated in Table 4. In the result section of the manuscript, the following was added.

Line 161-165

The PedKTAS level 1 in the trauma group had the highest ICU admission rate at 29.5%. Regarding medical diseases, ICU admission rates were the highest in the cardiac disease group at PedKTAS level 1 and level 2, at 26.4% and 14.4%, respectively. The results of hospitalization rate and ICU admission rate according to PedKTAS level for each disease group are shown in Table 4.

Table 4. Regression analysis according to triage levels by disease group of patients visiting pediatric emergency department

Variables

KTAS 1

KTAS 2

KTAS 3

KTAS 4

KTAS 5

Infectious disease

N=185

N=23357

N=325241

N=423318

N=37630

   Hospitalization (%)

85 (46.0)

11948 (51.2)

45326 (13.9)

37426 (8.8)

2435 (6.5)

      OR (95% CI)

12.47 (9.28-16.76)

10.41 (9.91-10.94)

2.37 (2.27-2.47)

1.51 (1.44-1.57)

Reference

      P value

<0.001

<0.001

<0.001

<0.001

   ICU admission (%)

23 (12.4)

1289 (5.5)

502 (0.2)

368 (0.1)

52 (0.1)

      OR (95% CI)

142.7 (77.45-262.95)

11.51 (8.71 - 15.22)

1.14 (0.86 - 1.52)

0.99 (0.74 -1.33)

Reference

      P value

<0.001

<0.001

0.3569

0.9579

Respiratory disease

N=827

N=45281

N=585874

N=581262

N=78493

   Hospitalization (%)

569 (68.8)

14100 (31.1)

88744 (15.2)

56629 (9.7)

7212 (9.2)

      OR (95% CI)

21.92 (18.87-25.47)

3.89 (3.77-4.02)

1.77 (1.72-1.81)

1.14 (1.11-1.17)

Reference

      P value

<0.001

<0.001

<0.001

<0.001

   ICU admission (%)

145 (12.4)

920 (2.0)

754 (0.1)

219 (0.1)

64 (0.1)

      OR (95% CI)

296.27 (216.99-404.51)

17.69 (13.68-22.87)

1.75 (1.36-2.27)

0.58 (0.44-0.77)

Reference

      P value

<0.001

<0.001

<0.001

<0.001

Gastrointestinal disease

N=38

N=2810

N=81357

N=148683

N=16527

   Hospitalization (%)

15 (39.5)

922 (32.8)

22732 (27.9)

17902 (12.0)

766 (4.6)

      OR (95% CI)

11.29 (5.77-22.08)

8.49 (7.61-9.46)

7.08 (6.57-7.63)

2.59 (2.41-2.79)

Reference

      P value

<0.001

<0.001

<0.001

<0.001

   ICU admission (%)

4 (10.5)

173 (6.2)

171 (0.2)

81 (0.1)

7 (0.1)

      OR (95% CI)

359.39 (92.91-1390.14)

65.75 (30.76-140.54)

6.77 (3.17-14.46)

2.03 (0.94-4.41)

Reference

      P value

<0.001

<0.001

<0.001

0.073

Neurological disease

N=2730

N=5250

20839

N=13112

N=1248

   Hospitalization (%)

2062 (75.5)

3322 (63.3)

10066 (48.3)

4869 (37.1)

297 (23.8)

      OR (95% CI)

9.7 (8.27-11.38)

5.09 (4.41-5.88)

3.27 (2.85-3.74)

2.16 (1.88-2.48)

Reference

      P value

<0.001

<0.001

<0.001

<0.001

   ICU admission (%)

384 (14.1)

363 (6.9)

293 (1.4)

38 (0.3)

9 (0.7)

      OR (95% CI)

22.2 (11.4-43.21)

9.2 (4.73-17.88)

2.09 (1.07-4.06)

0.46 (0.22-0.95)

Reference

      P value

<0.001

<0.001

0.0306

0.0347

Cardiac disease

N=1279

N=1443

N=8165

N=6602

N=447

   Hospitalization (%)

551 (43.1)

687 (47.6)

2464 (30.2)

1388 (21.0)

112 (25.1)

      OR (95% CI)

1.73 (1.35-2.21)

2.71 (2.13-3.44)

1.45 (1.16-1.81)

0.91 (0.73-1.14)

Reference

      P value

<0.001

<0.001

<0.001

0.3996

   ICU admission (%)

338 (26.4)

208 (14.4)

246 (3.0)

41 (0.6)

7 (1.6)

      OR (95% CI)

15.64 (7.31-33.48)

10.21 (4.76-21.9)

2.28 (1.07-4.88)

0.48 (0.21-1.08)

Reference

      P value

<0.001

<0.001

0.0336

0.0757

Trauma

N=579

N=20057

N=225811

N=1194108

N=135774

   Hospitalization (%)

356 (61.5)

4746 (23.7)

24318 (10.8)

26468 (2.2)

1836 (1.4)

      OR (95% CI)

117.61 (98.38-140.59)

23.72 (22.41-25.12)

8 (7.63-8.40)

1.63 (1.56-1.71)

Reference

      P value

<0.001

<0.001

<0.001

<0.001

   ICU admission (%)

171 (29.5)

902 (4.5)

884 (0.4)

562 (0.1)

31 (0.1)

      OR (95% CI)

1717.91 (1156.06-2552.83)

194.97 (136.27-278.96)

15.82 (11.06-22.62)

2.08 (1.45-2.99)

Reference

      P value

<0.001

<0.001

<0.001

<0.001

Values are presented as n (%) or odds ratio (95% confidence interval).
*P values are based on multiple logistic regression compared with KTAS 5.
KTAS: Korean triage and acuity scale, ICU: intensive care unit

-please provide data on severity outcomes such as mortality, length of stay

Response: Thank you for your important note. Mortality according to PedKTAS level in the PED are shown in Table 3. Data on emergency department length of stay according to Ped KTAS level were added to Table 3, and the Result section of the manuscript was added as follows.

Line 151-155

Deaths in the PED were 0.6% in 75 patients at PedKTAS level 1, and 47 patients and 29 patients and 9 patients at Ped KTAS levels 2, 3, and 4, respectively. The emergency department length of stay (LOS) was the longest at PedKTAS level 1 at 268.8 minutes, and the emergency department LOS decreased from PedKTAS level 1 to level 5 (Table3).

Discussion

-Authors compared the hospitalization and ICU rates based on PedKTAS levels, with those of the PedCTAS levels. Did the authors check if the study populations were comparable?

Response: Thank you for your important note. The studies shown in Figure 2 vary considerably in the number of subjects. A total of 550,940 patients were analyzed in the PedCTAS multicenter study, targeting all pediatric patients under the age of 18 who visited 12 pediatric emergency rooms, whereas our PedKTAS study analyzed a total of 5,462,964 pediatric patients under the age of 15 who visited all emergency rooms in Korea was analyzed. However, we did not statistically compare the PedKTAS-based hospital admission rate and ICU admission rate with those of PedCTAS, but simply showed the ratio of hospital admission rate and ICU admission rate. A follow-up study is planned to interpret the difference in results according to the level of PedKTAS and pedCTAS.

-Authors reported lower rates of hospitalization and ICU admissions due to the trend of over-triage of the PedKTAS tool. They argued that pediatric patients visit PEDs with fear, their irritability worsens and their condition may be evaluated as more severe than it actually is. What did the authors suggest to face those situations that are relatively frequent? Did the authors mean that PedKTAS is not sufficiently acute to triage those patients?

Response: Thank you for your very incisive point. Among the evaluation items of PedKTAS, there are two items in which the subjective evaluation of the nurse can intervene: the degree of pain in children and whether the child looks sick. Children's pain level is evaluated by NRS (Numeral Rating Scale) for children who can speak, and FPRS (Faces Pain Rating Scale) for children who cannot speak or are less than 3 years of age. However, when children visiting the emergency department are evaluated with a tool such as FPRS because of fear of hospitals and medical staff, the level of pain is measured high, so the severity can be measured high. In addition, in the evaluation of whether the child looks sick, if the child cries or is afraid, the child is evaluated as looking sick, which can lead to an increase in severity. This may be a limitation of PedKTAS. Therefore, the authors of this study are contemplating ways to minimize the impact on these subjective evaluation items. One of the methods is to evaluate with modified PedKTAS excluding subjective evaluation items. Then, by comparing and analyzing the results with those of PedKTAS, the performance of modified PedKTAS is identified. Follow-up studies will be needed to create a triage tool that can sufficiently screen emergency pediatric patients without these subjective evaluation items.

-In the future, will the authors plan to analyze the validity of their triage tool PedKTAS versus a gold standard? (i.e. PedCTAS)

Response: Thank you for your important note. In this study, hospitalization rates and ICU admission rates of PedKTAS and PedCTAS were simply compared. However, we plan to carry out a statistical comparative analysis according to each triage level in the future. If there is a difference in the analysis results, we will identify the cause, and modify and supplement PedKTAS so that pediatric emergency patients can be detected early and treated promptly.

Reviewer 2 Report

dear Editor, I read with interest the article you sent me for my review. The Authors show interesting retrospective data on one of the main tools available to a pediatric emergency department, which is triage system.

I'm not a native English speaker so I limit myself to a few lexical suggestions, leaving the revision of the language to more competent people.

However, I have some concerns about the following concepts that I report below:

Introduction

The PedCTAS is one of the main triage tools used worldwide, and my department has also introduced it for about two years with excellent results in terms of risk stratification and patient waiting time; the Authors should better explain what are the reasons that led to the creation of a new triage system, i.e. the PedKTAS. I would also suggest reporting a summary image/table in which the two triage systems are compared, thus making it more usable when reading the discussion with the comparison between the results applying the two different systems.

Materials and Methods

line 72: I would suggest to rewrite the "< 0 years" with "< 1 years", because it would be clearer; alternatively, authors can write 0-1 years.

lines 89-90: I suggest to rewrite the sentence "The secondary outcome was to compare the hospitalization and ICU admission rates, based on the PedKTAS levels, with those of the PedCTAS levels" e.g. "The secondary outcome was to compare the hospitalization and ICU admission rates with application of PedKTAS and PedCTAS, respectively"

Results

lines 101-102. the Authors excluded approximately 1/5 of the patients from the study; even if the remaining patients included are surely a sufficient number for an accurate statistical analysis, such a high exclusion number may represent a bias; the Authors should therefore better explain which are the missing data that led to the exclusion of the patients and report them within the limits of the study.

Discussion and conclusion.

One of the main results of the study is that by applying PedKTAS there are fewer admissions to the ICU than by applying PedCTAS, which could mean that in the leve 1 PedKTAS triage there is an overestimation of severity; even if it is always better to overestimate rather than underestimate the severity of a pediatric patient, I believe that the Authors should stressed more this point, suggesting that PedKTAS is less permissive than PedCTAS, with a higher number of level 1 but with a lower rate of hospitalization in the ICU; this has considerable importance from a clinical point of view, because it increases the speed with which the grade 1-2 patient accesses treatment and therefore a more rapid management of the patient.

I believe that the Authors should underline the importance of the topic and the need ofcontinuing with the scientific research, stimulating other Authors to provide contributions in this field.

Minor recommendations/revisions: an interesting data, which I don't know if it is available to the authors, could also be the degree of re-entries after 48-72h of discharged patients, to actually see if the first degree of triage assigned was correct.

another interesting data could be the subdivision of the ICU admission rate with comparison between PedKTAS and PedCTAS divided by age groups that identified the authors, to verify if, as usually happens, there is a greater overestimation of the severity in patients smaller.

Author Response

Thank you for taking the time to review our manuscript and for providing meaningful comments. We have made corrections and clarifications in the revised manuscript after carefully reading your comments.

Introduction

The PedCTAS is one of the main triage tools used worldwide, and my department has also introduced it for about two years with excellent results in terms of risk stratification and patient waiting time; the Authors should better explain what are the reasons that led to the creation of a new triage system, i.e. the PedKTAS. I would also suggest reporting a summary image/table in which the two triage systems are compared, thus making it more usable when reading the discussion with the comparison between the results applying the two different systems.

Response: Thank you for your important note. Since I did not clearly describe the reason for choosing PedKTAS in Korea, I think it is something that readers might be curious about. This is explained in more detail in the Introduction section as follows. In addition, we have made supplementary table 2 so that readers can more easily understand the contents of the comparison between PedKTAS and PedCTAS.

Line 44-57

Before 2015, different triage systems were applied to children in each emergency department nationwide. Each emergency department classified patients' urgency into three levels by applying ESI, classified into five levels by applying PedCTAS, and even some emergency departments did not apply the Triage system. For this reason, it was difficult to share information about patients' emergency levels between hospitals or between 119 emergency medical services and medical staff at the pre-hospital stage. As a result, in 2010, an incident occurred in which a child with intussusception died because the emergency department of several hospitals did not recognize the emergent situation of the patient. Given these circumstances, a nationally unified triage system was needed to evaluate the degree of urgency in pediatric emergency patients. The Korean Society of Emergency Medicine (KSEM) developed the Pediatric Korean Triage and Acuity Scale (PedKTAS) [6] in 2015 by modifying and supplementing the PedCTAS to suit the situation in Korea because PedCTAS was known as a triage tool whose reliability and validity were verified.

Line 115-117

There are differences between the two triage tools in terms of application age, chief complaint list, and criteria for abnormal vital signs. Details of the comparison between PedKTAS and PedCTAS are described in Table 1.

Table 1. Comparison of  Pediatric Korean Triage and Acuity Scale and Pediatric Canadian Triage and Acuity Scale characteristics

Criteria

PedKTAS

PedCTAS

Time to triage assessment

Immediately

10 min

Applicable age

< 15 years old

< 18 years old

Time to nurse reassessment

Immediate/15/30/60/120 min

Immediate/15/30/60/120 min

Triage decision criteria

Complaints based with 1st/2nd
modifiers used

Complaints based with 1st/2nd
modifiers used

Chief complaint list

17 categories (166 chief complaints)

18 categories (161 chief complaints)

Criteria for vital signs

Further segmentation by age (every 3 months under 3 years old, every year over 3 years old)

Organized by age group (every 3 months for children under 1 year old, 1-3 years old, 4-6 years old, 7-10 years old)

Range of abnormal vital signs

Wider

Narrower

Education method

Online or offline training

Web-based training

PedKTAS: Pediatric Korean triage and acuity scale, PedCTAS: Pediatric Canadian Triage and Acuity Scale.

Materials and Methods

line 72: I would suggest to rewrite the "< 0 years" with "< 1 years", because it would be clearer; alternatively, authors can write 0-1 years.

Response: Thank you for pointing this out. It's my mistake. It will be revised to “< 1 years".

Line 90-92

We divided the participants into four age groups: <1 year (i.e., infant), 1–4 years (i.e., young child), 5–9 years (i.e., preschool- and early school-aged child), and 10–14 years.

lines 89-90: I suggest to rewrite the sentence "The secondary outcome was to compare the hospitalization and ICU admission rates, based on the PedKTAS levels, with those of the PedCTAS levels" e.g. "The secondary outcome was to compare the hospitalization and ICU admission rates with application of PedKTAS and PedCTAS, respectively"

Response: Thank you for pointing this out. As you said, I will revise it to convey the meaning more accurately.

Line 120-121

The secondary outcome was to compare the hospitalization and ICU admission rates with application of PedKTAS and PedCTAS, respectively.

Results

lines 101-102. the Authors excluded approximately 1/5 of the patients from the study; even if the remaining patients included are surely a sufficient number for an accurate statistical analysis, such a high exclusion number may represent a bias; the Authors should therefore better explain which are the missing data that led to the exclusion of the patients and report them within the limits of the study.

Response: Thank you for your important note of what we were missing. Patients younger than 15 years of age who visited the PED were included in this study. 12,822 patients with CTAS 8 or 9 who visited the PED for certificate issuance and 1,309,867 patients not classified as PedKTAS were excluded from this study. Among the patients not classified as PedKTAS, most of them (1,286,812) were patients who visited Regional EMI. This is probably due to the fact that many regional EMIs did not follow the KTAS classification guidelines or did not transmit KTAS to NEDIS in the early days of nationwide implementation of KTAS. In fact, when checked by year, the number of patients missing KTAS is decreasing from 518,192 in 2016, 392,721 in 2017, 303,847 in 2018, and 72,052 in 2019. This is a limitation of this study and was added as follows to the method and discussion section.

Line 84-86

Patients younger than 15 years of age who visited the emergency department were included. Patients with KTAS 8 and 9 who were not classified as PedKTAS or who visited a PED for certificate issuance were excluded from the analysis.

Line 282-288

Third, 1,309,867 patients not classified as PedKTAS were excluded from this study. Most of these patients visited Regional EMI, about one-fifth of the total number of pediatric patients. This was confirmed to have occurred in the early days of implementing KTAS nationwide. Due to this, there is a possibility that sample selection bias exists. In the future, it is thought that additional analysis will be needed by extending the study period except for the initial period of KTAS implementation.

Discussion and conclusion.

One of the main results of the study is that by applying PedKTAS there are fewer admissions to the ICU than by applying PedCTAS, which could mean that in the leve 1 PedKTAS triage there is an overestimation of severity; even if it is always better to overestimate rather than underestimate the severity of a pediatric patient, I believe that the Authors should stressed more this point, suggesting that PedKTAS is less permissive than PedCTAS, with a higher number of level 1 but with a lower rate of hospitalization in the ICU; this has considerable importance from a clinical point of view, because it increases the speed with which the grade 1-2 patient accesses treatment and therefore a more rapid management of the patient.

Response: Thanks for pointing out something I hadn't thought of. The fact that the ICU admission rate of PedKTAS level 1 patients is lower than that of PedCTAS level 1 patients has important clinical significance. I think further explanation is needed on this. The following explanation has been added to the Discussion section.

Line 247-254

The fact that the ICU admission rate of PedKTAS level 1 patients is lower than that of PedCTAS level 1 patients has important clinical significance. PedKTAS may be overestimating the severity and urgency of pediatric patients. It may be better to overestimate than underestimate the condition of pediatric patients. However, in places where the use of medical resources is limited, such as an emergency department, too many Level 1 patients requiring prompt treatment may be screened, and treatment of patients actually requiring emergency treatment may be delayed. Therefore, it may be necessary to modify the evaluation items for selecting PedKTAS lever 1 patients.

I believe that the Authors should underline the importance of the topic and the need of continuing with the scientific research, stimulating other Authors to provide contributions in this field.

Response: Thank you for your important note. The screening of emergency patients is a factor that directly affects patient safety due to overcrowding of emergency departments. This is a very important issue that many researchers and clinicians will continue to study. The following explanation has been added to the end of the Discussion section.

Line 291-301

The emergency department has the characteristics that the number of patients visited and the condition cannot be predicted, and treatment must be performed with limited medical resources. Therefore, overcrowding in emergency departments is an important problem that is directly related to the threat to patient safety. Efforts are being made in various ways to relieve the overcrowding of emergency departments. Using the triage system, the flow of patients can be controlled in the pre-hospital stage, and the patient's condition can be accurately evaluated at the initial stage so that patients visiting the emergency department can wait safely in the hospital stage. Through this, overcrowding in the emergency department can be improved. Continuous efforts and research of many researchers will be required for the development and performance improvement of the triage system.

Minor recommendations/revisions: an interesting data, which I don't know if it is available to the authors, could also be the degree of re-entries after 48-72h of discharged patients, to actually see if the first degree of triage assigned was correct.

Response: Thank you for your important note. Unfortunately, because this study used anonymized data, it is not possible to know if the patient returned to the hospital. In the future, it will be possible to check whether the triage evaluated at the first visit in returning patients was appropriate using data from individual institutions that were not anonymized. The authors of this study are conducting a follow-up study on this.

another interesting data could be the subdivision of the ICU admission rate with comparison between PedKTAS and PedCTAS divided by age groups that identified the authors, to verify if, as usually happens, there is a greater overestimation of the severity in patients smaller.

Response: Thank you for your kind comments. Unfortunately, age was not presented in the PedCTAS study compared to our study, so subgroup analysis was not possible. However, the younger the patient, the more likely the patient's condition will be overestimated, and comparative analysis with PedCTAS is thought to be meaningful. According to you, we will plan a follow-up study on this.

Round 2

Reviewer 1 Report

Thank you for all your revisions.

Although the authors have well described the implementation of PedKTAS, would it be possible to translate the PedKTAS tool into english?(appendix A)

Because the PedKTAS appears to be a variant of the PedCTAS, it would be valuable to provide readers with a better understanding of their pediatric triage tool

Author Response

Although the authors have well described the implementation of PedKTAS, would it be possible to translate the PedKTAS tool into english?(appendix A)

Because the PedKTAS appears to be a variant of the PedCTAS, it would be valuable to provide readers with a better understanding of their pediatric triage tool

Response: Thank you for the quick review. I agree with what you are saying. I think it will be helpful for readers to understand the screen shot (Appendix A) of implementing PedKTAS in English. Appendix A was uploaded in English.
